# Is there a direct solar proton impact on lower stratospheric ozone?

Jia Jia[1], Antti Kero[1], Niilo Kalakoski[2], Monika E. Szeląg[2*], Pekka T. Verronen[1,2]

[1] Sodankylä Geophysical Observatory, University of Oulu, Sodankylä, Finland
[2] Space and Earth Observation Centre, Finnish Meteorological Institute, Helsinki, Finland
[*] earlier known as M. E. Andersson

*Correspondence to*: Jia Jia (jia.jia@oulu.fi)

**Abstract.** We investigate Arctic polar atmospheric ozone responses to Solar Proton Events (SPEs) using MLS satellite measurements (2004–now) and WACCM-D simulations (1989–2012). Special focus is on lower stratospheric (10–30 km) ozone depletion that has been proposed earlier based on superposed epoch analysis (SEA) of ozonesonde anomalies (up to 10% ozone decrease at ~20 km). SEA of the satellite dataset provides no solid evidence of any average SPE impact on the lower stratospheric ozone, although at the mesospheric altitudes a statistically significant ozone depletion is present. In the individual case studies, we find only one potential case (January 2005) in which the lower stratospheric ozone level was significantly decreased after the SPE onset (in both model simulation and MLS observation data). However, similar decreases could not be identified in other SPEs of similar or larger magnitude. Due to the input proton energy threshold of > 300 MeV, the WACCM-D model can only detect direct proton effects above 25 km, and simulation results before the Aura MLS era indicate no significant effect on the lower stratospheric ozone. However, we find a very good overall consistency between WACCM-D simulations and MLS observations of SPE-driven ozone anomalies, both on average and for the individual cases including January 2005.

# 1 Introduction

In the near-Earth space, solar wind charged particles are guided by the Earth's magnetic field and are able to precipitate into the middle and upper atmosphere in the polar regions. Such kind of precipitation creates the spectacular aurora, but also produces considerable amounts of $HO_x$ (H, OH, $HO_2$) and $NO_x$ (N, NO, $NO_2$) through ion-neutral chemistry (e.g. Verronen and Lehmann, 2013). $HO_x$ and $NO_x$ increases lead to ozone loss through catalytic reactions in the mesosphere and upper stratosphere, respectively (Sinnhuber et al., 2012). Moreover, in polar winter, $NO_x$ has a long chemical lifetime due to limited photodissociation by solar radiation. $NO_x$ produced by energetic particle precipitation (EPP) in the mesosphere-lower thermosphere is transported down to the stratosphere by the Brewer-Dobson circulation inside the polar vortex (Funke et al., 2014), causing depletion of upper stratospheric ozone (Damiani et al., 2016). A number of studies have confirmed EPP's remarkable role in ozone depletion directly during large EPP events (e.g. Funke et al., 2011) and indirectly due to descending $NO_x$ (e.g., Randall et al., 2007). Thus, many advanced chemistry-climate models are now including EPP forcing, in order to correctly represent the ozone distribution in the polar stratosphere and mesosphere (Matthes et al., 2017; Stone et al., 2018).

Solar proton events (SPE) are one of the main types of EPP. During SPE, particles (mainly protons) with energies from tens to hundreds of MeV precipitate into the atmosphere at geomagnetic latitudes larger than 60° for days. Such high-energy particles mainly affect the atmosphere at altitudes of 35–90 km, providing direct ionization forcing on the polar middle atmosphere. Large SPEs have been studied since the 1960s until today using satellite observations and model simulation. In addition to tens of percent of ozone loss observed at altitudes above 35 km (Jackman 2001, Seppälä et al. 2004, Verronen et al. 2006), a strong SPE can reduce total ozone by 1–3% for months after the event (Jackman 2011, 2014).

Recently, Denton et al. (2018 a, b) presented statistical studies of average ozone changes from 191 SPEs between 1989–2016 using ozonesonde measurements. Superposed epoch analysis of ozone anomalies at polar stations (Sodankylä, Ny-Ålesund, and Lerwick) indicated that SPEs occurring during winter are causing ozone decrease by 5–10%, on average, at 20 km altitude. This effect is not produced in the current models because SPE-induced ionisation rates are insignificant at this altitude even during largest events with high proton energies from 300–20000 MeV (Jackman et al., 2011). Denton et al. (2018 a, b) included also a large number of very small SPEs in their analysis. Such ozone decreases have not been observed in the case studies of very extreme (particles with energies >10 MeV are greater than 10 000 particle flux units) SPEs, e.g., the 2003 'Halloween' event, from either simulation or satellite observation (Funke et al. 2011 and references therein). Recently, statistical analysis based on simulations has found no evidence of such low-altitude ozone impact (Kalakoski et al. 2020). Moreover, from the chemical aspect, we also rather expect ozone increase at lower stratosphere due to the enhanced NOx interfering with chlorine-driven catalytic ozone loss (Jackman et al., 2008).

Here we investigate the proposed SPE-induced direct depletion on lower stratospheric (10–30 km) ozone using ozone data from the Microwave Limb Sounder (MLS) instrument aboard the Aura satellite and the Whole Atmosphere Community Climate Model (WACCM-D) simulations. We proceed to evaluate ozone changes at altitudes 10–70 km caused by SPEs both statistically (superposed epoch analysis) and individually (case by case). The MLS ozone data, WACCM-D atmospheric simulation, and SPE data sets are presented in Sect. 2. In order to

cross-check ozone depletion at 20 km reported based on the ozonesonde data, statistical ozone responses from MLS satellite measurements are firstly provided in Sect. 3. Following that, MLS and WACCM-D ozone changes after individual SPEs are given in Sect. 4. Finally, we summarize our results and conclusions in Sect. 5.

## 2    Data sets

### 2.1    O$_3$ profile measurements by MLS

MLS onboard the Earth Observing System (EOS) Aura satellite measures ozone emission at 240 GHz, providing ozone volume mixing ratios at 55 pressure levels since 15 July 2004 (Waters et al., 2006). Vertical profiles are retrieved from the MLS observations with a 165 km horizontal spacing at altitudes between 8 and 90 km, a spatial resolution of ~400 km horizontal and ~3.2 km vertical. In this work, we use version 4.2 ozone data measured at 261–0.02 hPa (~10–70 km) to calculate the daily averaged ozone density profile at northern high latitudes (60°–90°N). Readers who are interested in the MLS data quality are referred to Livesey et al. (2018).

### 2.2    O$_3$ from WACCM-D simulations

WACCM is a global circulation model, including fully coupled dynamics and chemistry. Here, we use version 4 of the WACCM with resolution of 1.9° latitude by 2.5° longitude, with 88 vertical levels reaching from surface to $6\times10^{-6}$ hPa ($\approx$140 km). Overview of the model and the description of climate and variability in long-term simulation was presented by Marsh et al. (2013), with details of model physics in MLT (mesosphere - lower thermosphere region) and the response of the model to radiative and geomagnetic forcing during solar maximum and minimum described by Marsh et al. (2007). The simulation results presented here are from WACCM-D, a variant of WACCM with more detailed set of lower ionospheric chemical reactions, aimed at better reproduction of observed effects of EPP on MLT neutral composition (Verronen et al., 2016; Andersson et al., 2016).

We use SD-WACCM-D specified dynamics configuration, with Modern-Era Retrospective Analysis for Research and Applications (MERRA) (Rienecker et al., 2011) meteorological fields to force dynamics at every time step up to about 50 km. Simulation covers years 1989-2012, and uses forcings from auroral electrons (E<10keV), solar protons (E<300 MeV), and galactic cosmic rays for energetic particle precipitation. The SPE ionization rates are based on proton flux measurements from the Geostationary Operational Environmental Satellites (GOES) (see e.g. Jackman et al., 2011, for the calculation method). The WACCM-D SPE effects on neutral species are compared to satellite observations in Andersson et al. (2016). Note that WACCM-D has not been validated below 20 km. Nevertheless, in Andersson et al. (2016) the HNO$_3$ response above 15 km to single SPE onset was reasonable compared to MLS data. We also stress that protons with energy over 300 MeV are not included in the simulation. 300 MeV protons mostly affect the atmosphere at around 25 km (Turunen et al., 2009; Wissing and Kallenrode, 2009). As 300 MeV is the upper limit of the proton energies considered in our model simulation, the WACCM-D simulation presented here can therefore only investigate the impact of direct proton forcing at altitudes above 25 km. For more details of the simulation setup, see Kalakoski et al. (2020).

## 2.3 Solar proton events

The data of solar proton events (SPEs) used in this study is based on NOAA GOES proton flux observations. Fig.

1 presents 261 SPEs recorded from 1975 to date, including their onset time, fluxes detected in space, approximated

time of duration, and average ionization rates to the atmosphere at two altitudes. Here, the onset of a SPE is

defined as the time when 5-min average proton fluxes with energies >10MeV are greater than 10 Particle Flux

Units (1 pfu = 1 particle /cm$^2$/s/sr) at the geosynchronous orbit. For the estimation of SPE duration and its impact

on the atmosphere, we use the daily average ion pair production rates at ~1 hPa (~46 km, upper panel) and ~12

8 hPa (~29 km, lower panel). These ionization rates are calculated from GOES proton flux observations using the

9 energy deposition methodology described in, e.g., Jackman et al. (2011). The SPE durations presented here were

10 calculated as the period when the ionization rates at ~1 hPa / 12 hPa are larger than 2 ion pair/cm$^3$/s before the

11 next event starts. The average ionization rates in Fig. 1 were then derived by averaging the ionization rates at 1

12 hPa / 12 hPa during this period. Our study used 49 events that occurred after the launch of Aura MLS (July 2004–

13 now) and 177 events that occurred in the complete WACCM-D simulation period (Jan 1989–Dec 2012) to

14 evaluate the ozone changes following SPEs. It is clearly demonstrated in Fig. 1 that these SPEs are more frequent

near solar maximum years. Majority of the events are with flux less than 400 pfu, and their impacts to the

atmosphere below 1 hPa are small. It is worth to mention that although these SPEs seem to have no preference in

occurring season, their seasonal distribution varies by months and should be considered during the interpretation

(Fig. A1).

19 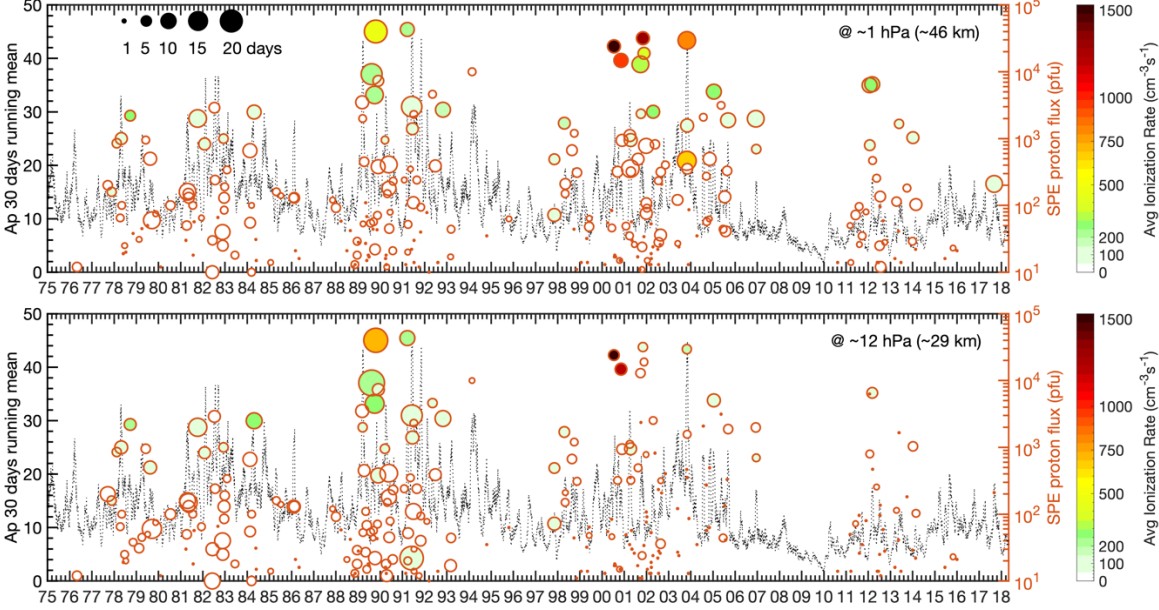

**Figure 1. Onset time of SPEs and their proton fluxes since 1975. The filled colors are the average ionization rate during**

**each SPE at ~1 hPa (upper panel) and ~12 hPa (lower panel), while the size of the markers represents the approximate**

**duration time of the SPEs obtained from the daily mean ionization rate at the two altitudes. The black dotted line in**

**the background is the 30-day mean of the daily geomagnetic activity Ap-index.**

## 3    Statistical O₃ response from MLS

Similar to the method used by Denton et al. (2018 b), we applied a superposed epoch analysis to the MLS daily ozone anomalies. The superposed epoch analysis, also referred to as composite analysis in geophysics, is used to acquire variation of a time series before and after an event or a chain of certain kind of events. The point of time when the event begins is the epoch time. In this case, the epoch times are the onset times of individual SPEs during MLS operating period. All available ozone data were binned as a function of epoch time and altitude, with temporal resolution of one day. The pre- and post-epoch spans used here are 30 and 60 days, respectively. For the selected sets of SPEs, all the binned ozone data sets were averaged to represent the effect of the SPEs. This method excludes natural ozone variations that are larger than the span-scale. Since SPE-driven effects are expected to take place on daily to monthly time scales, variations caused by e.g. QBO can be excluded. However, seasonal variations must be excluded before using superposed epochs. Thus, the daily profile climatology calculated from the ozone data was subtracted from the daily ozone data. Different from Denton et al. (2018 a, b), to make sure SPEs are 'isolated' from the previous events, events that happened within 10 days of the previous SPE were excluded.

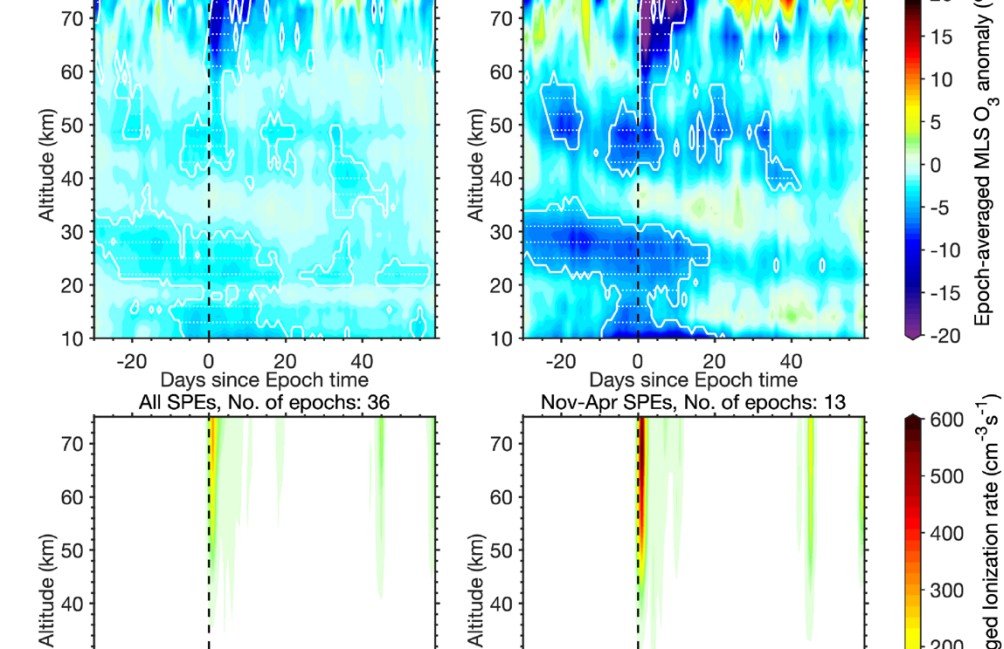

**Figure 2. Epoch-averaged MLS ozone anomalies (relative in %) (upper panels) and the corresponding daily ionization rates (lower panels) in the northern polar region (60°-90°N) along with geopotential altitude for a total of 35 'isolated' SPE epochs (left panel) and 13 'isolated' winter SPE epochs (right panel). The black dashed line represents the epoch time, i.e., onset of SPEs. The white thick line area corresponds to the epoch-averaged anomalies with >95% confidence after the Monte Carlo test.**

In order to test the statistical significance of the obtained results, a Monte Carlo test was implemented. Instead of using SPE onset as epoch times, the analysis was rerun using 2000 random sets of epoch times. SPE-epoch averaged variations larger than 95% of the 2000 randomized results are considered significant and robust (reported as >95% confidence), suggesting that these extracted signatures are likely not random, but related to SPE or driven by some other external forcing.

Fig. 2 shows the superposed epoch of MLS northern polar ozone anomalies and the corresponding daily ionization rates for all 'isolated' SPEs (35 out of 49 events, left panels) and for the ones occurring in winter (Nov-Apr) (13 out of 19 events, right panels) within the instrument's operational period. Robust averaged anomalies (>95% confidence) are presented within the white thick lines. Spatial distribution of statistically robust anomalies is similar in all-SPE epochs and winter-SPE epochs. The depletion is more pronounced for winter epochs. This, of course, could be a statistical effect due to the much lower number of events used in the study, but is also expected due to two facts: 1) ozone recovery is slower due to less production from $O_2$ photodissociation; 2) Largest SPEs with flux >1000 pfu that cause more ozone depletion happen to occur in NH winter. Among all the SPEs during MLS measurement period, ~3/4 of big SPEs are in NH wintertime (see Fig. A1). In both upper panels, closely following the SPE onset, very pronounced ozone depletion appears above 50 km for over 5 days. This is the direct ozone loss caused by the SPE-induced $HO_x$ enhancement. The number of extreme SPEs is relatively small, which explains the absence of the long-lasting ozone depletion that would be expected between 40–50 km from enhanced amounts of $NO_x$. While the upper stratospheric ozone depletion signature is not seen in the statistical average, 5–10% decrease of ozone is present below 30 km, including ozone loss around 20 km similar to that reported by Denton et al. (2018 a, b). However, since this variation starts already several days before the epoch, we cannot exclude the possibility that the whole robust variation in the stratosphere is more related to other phenomena in the northern polar cap, e.g. to changes in the strength of polar vortex or related chemical effects. We will discuss this in more detail in Sect. 4.

A superposed epoch analysis of WACCM-D ozone anomalies from SPEs during 1989–2012 has been reported by Kalakoski et al. (2020), thus we will not repeat it here. In their results, the epoch-averaged WACCM-D ozone anomalies showed the same robust depletion at above 50 km. Since their analysis included also the very large SPEs that occurred 1989–2004 (see Fig. 1 in this study, or list of largest 15 SPEs in Tab. 1, Jackman et al., 2008), long-term ozone depletion in the upper stratosphere was clearly detected as well. However, there was no robust ozone loss below 30 km found in the WACCM-D simulations.

**4    $O_3$ response to individual SPEs**

Considering the limited number of SPE events during MLS era, and the high variability of stratospheric ozone, influenced, e.g. by SSWs or heterogeneous chemistry on PSC surfaces, particularly during winter, in this section we analyse ozone responses to individual SPEs.

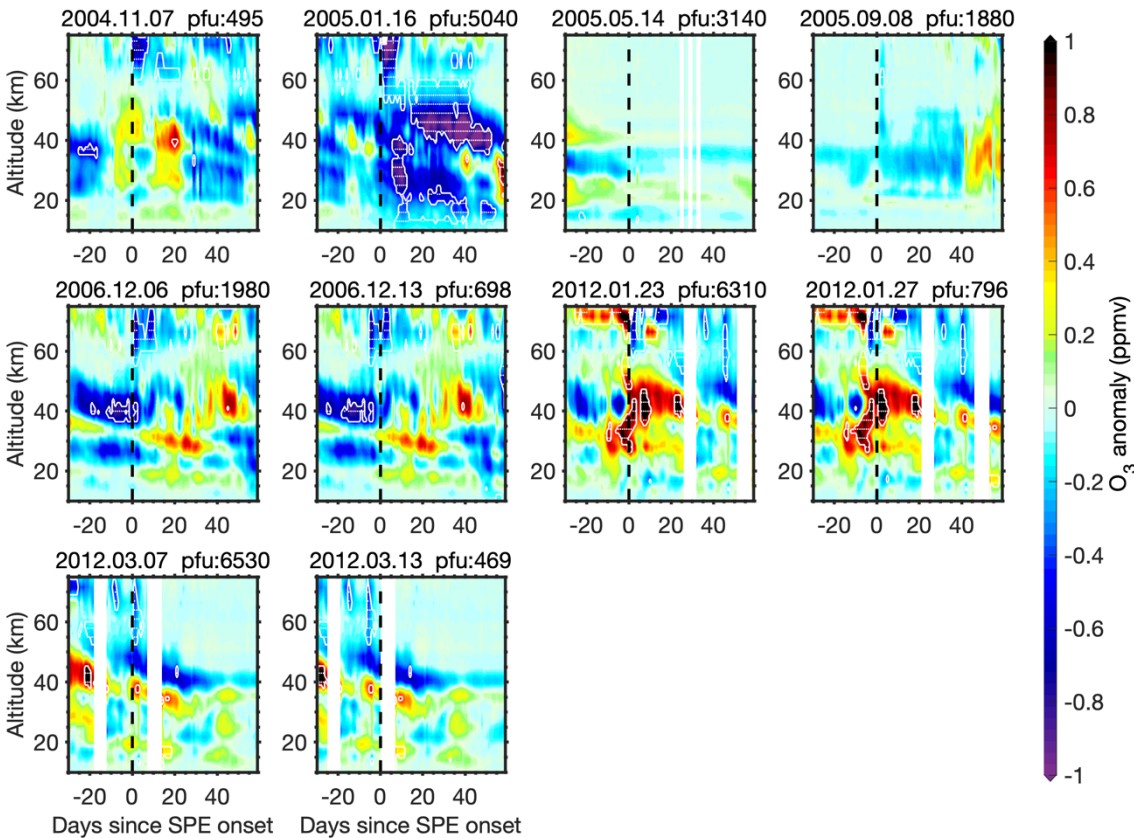

Figure 3. MLS ozone anomalies (in ppmv) along with altitude at 30 days before and 60 days after individual big SPEs (proton fluxes >400 pfu) in July 2004–December 2012. The white thick line area demonstrates ozone anomalies with >95% confidence after the Monte Carlo test.

Similar to the analysis presented in Sect. 3, ozone anomalies presented here were calculated by subtracting daily climatology from daily averaged ozone data from MLS and WACCM-D. For Figs 3 and 4, to make the results from MLS and WACCM-D simulation comparable, WACCM-D daily ozone was calculated using simulation profiles at MLS observation time and location. The climatology from MLS and WACCM-D were derived from their overlapping time period to guarantee a comparable background. For Figs. 6, A2 and A3, the subtracted daily mean climatology from MLS and WACCM-D were derived from the MLS data period and the WACCM-D simulation period, respectively. Then, instead of applying superposed epoch analysis on multiple SPEs, ozone anomalies are presented 30 days before and 60 days after onset of individual SPE. For estimating the statistical significance of the ozone anomalies found in the individual SPEs, we applied a similar Monte Carlo approach as in the case of SEA, i.e., the variance of 6000 random 'onset' times was used as a measure for a significant anomaly. It is worth noting, however, that this method recognizes all 'statistically significant' anomalies larger than the random background variation, whether the anomaly is due to SPE or, for instance, due to exceptional dynamical/chemical anomalies, which have a similar occurrence probability as SPEs.

Anomalies following all individual SPEs can be found in Figs. A2 and A3. In general, SPEs with proton fluxes < 400 pfu cause neither visible daily ozone depletion in the mesosphere (below 75 km), nor in other altitudes. Ozone

changes following individual SPEs are more pronounced during winter. Figs. 3 and 4 demonstrate MLS and WACCM-D ozone variations following SPEs with proton fluxes > 400 pfu in July 2004 – end of 2012. Both the ozone variations and the robust signatures from these two different data sets are very consistent. After 2004, three large winter SPEs, i.e., January 2005, September 2005 and March 2012, produced clear upper stratospheric ozone loss. Ozone depletion is most pronounced following the January 2005 event. For this event, we also observe a robust lower stratospheric ozone loss from MLS following SPE for the first time: ozone is depleted by ~1 ppmv (~15%) at 20–35 km and by ~0.15 ppmv (>20%) below 15 km 5 days after SPE onset.

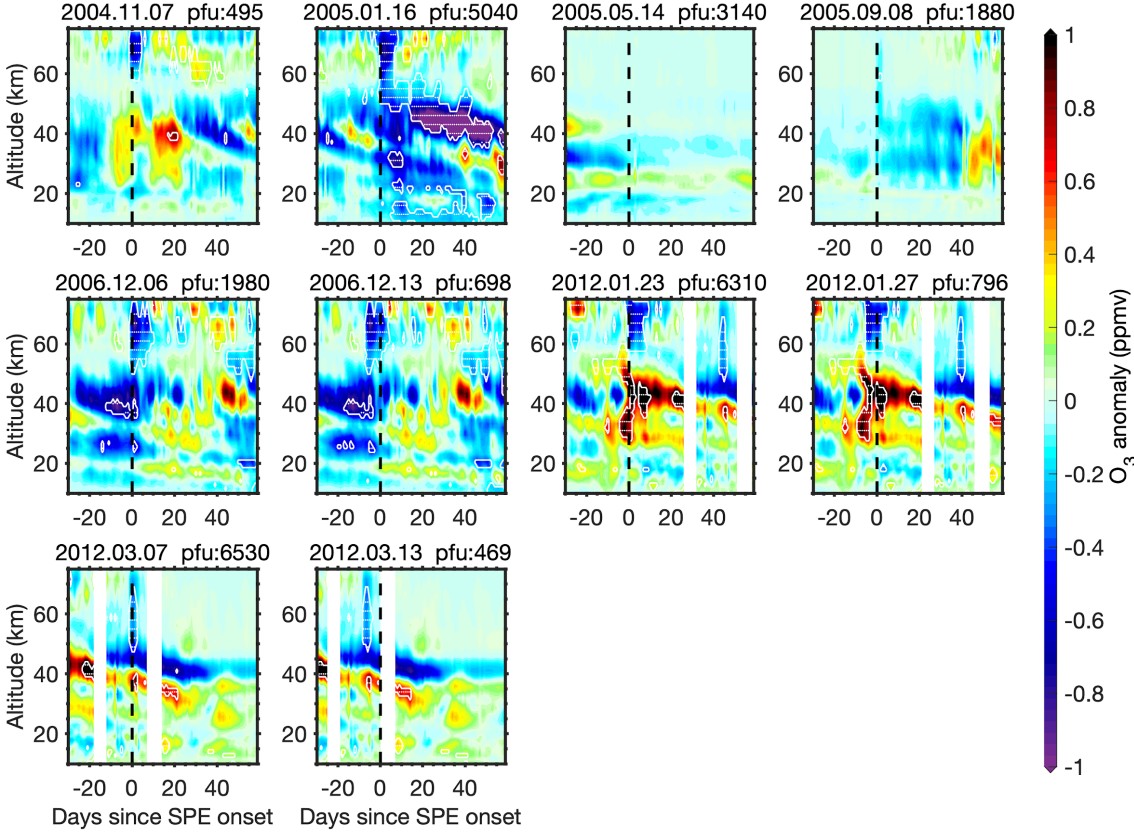

**Figure 4. Same as Fig. 3 but for ozone anomalies from WACCM-D simulation at MLS measurement time and location.**

Overall, wintertime ozone variation below 35 km is rather complicated. Year-to-year variability of stratospheric polar ozone is mostly controlled by dynamical and chemical processes, both are essentially coupled to temperature changes. Factors that modify polar temperature, e.g., sudden stratospheric warming (SSW) and El Niño–Southern Oscillation (ENSO), are essentially planetary wave perturbations that modulate the strength of polar vortex. The probabilities of major SSWs and, on the other hand, springs with extremely strong polar vortex are at similar levels as the one of SPEs. Thus, ozone variations by these events will be seen as robust signatures in our study as well, yet they do not necessarily coincide with onsets of SPEs with proton fluxes >400 pfu and >10000 pfu, respectively. The large SPE in January 2012 (Figs. 3 and 4) is severe enough to destroy stratospheric ozone. However, the stratospheric ozone anomalies at that time were dominated by dynamical ozone enhancement from SSW in 17th Jan 2012 (Päivärinta et al., 2016). One of the most pronounced examples of extreme strong polar

vortex impact is the well-reported ozone depletion during spring of 2011, which can be observed in ozone anom-

alies around the two small SPEs that occurred in March 2011 (see Fig. A2). The lower stratospheric polar vortex

was the strongest (in either hemisphere) in the previous 32 years (Manney et al., 2011). Large volume of polar

stratospheric clouds (PSCs) converted chlorine reservoirs to ozone-destroying species, leading to extraordinary

low ozone in the stratosphere (Pommereau et al., 2018). Similarly, robust anomaly seen after January 2016 SPE

can be explained by cold 2015–2016 winter anomaly. We are confident to exclude SPE's influence on the anomaly

in both cases because: firstly, the signal is not following SPE onset, secondly these SPEs are such small events

that ozone loss was not observed, not even in the mesosphere. These robust non-SPE signals are included in the

superposed-epoch analysis performed in Sect. 3, contributing to the robust anomalies below 30 km in Fig. 2.

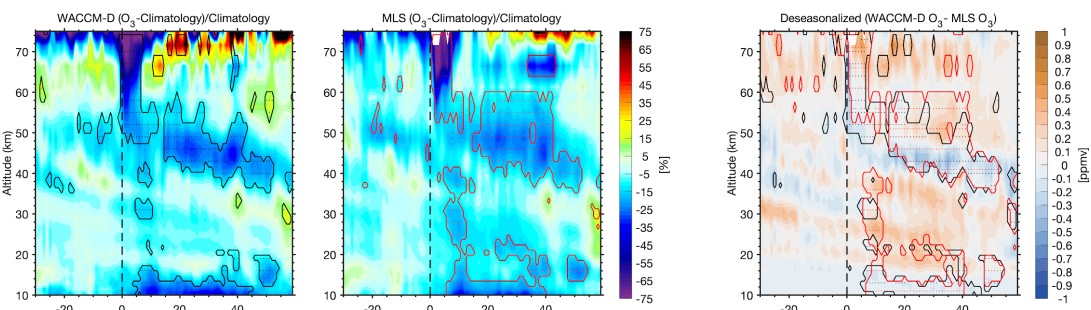

**Figure 5. WACCM-D (left panel) and MLS (middle panel) relative ozone anomalies along with altitude at 30 days**

**before and 60 days after SPE on 16th January 2005. The WACCM-D simulation used here are the profiles at MLS**

**measurement time and locations. The climatology was calculated using data between Jul 2004 – Dec 2012 for both MLS**

**and WACCM-D. The black/red thick line area demonstrates relative ozone anomalies with >95% confidence after the**

**Monte Carlo test. Right panel is ozone differences between WACCM-D and MLS during this time frame (de-**

**seasonalize means that the seasonal differences showed in Fig. S4 were removed). The black/red thick line area**

**demonstrates direct ozone anomalies with >95% confidence after the Monte Carlo test from WACCM-D and MLS**

**data, respectively.**

Identify sources of the robust ozone anomaly below 35 km following the SPE beginning on 16th Jan 2005 is

difficult. With a moderate cold winter temperature causing more ozone loss, coincident of robust dynamical ozone

changes following the SPE exists. Meanwhile, an extremely large (over 270%) ground level enhancement (GLE)

of neutrons occurred during the SPE period on 20 January 2005 (Jackman et al., 2011). Ionization rate reached

500 $cm^{-3}s^{-1}$ at 30 km for one day due to the very high energy protons (300–20 000 MeV) that caused the GLE

(Usoskin et al., 2011). Jackman et al. (2011) carried out a detailed study of January 2005 SPE's influence on the

northern polar atmosphere using WACCM3 simulation, and reported an ozone column decrease of less than 0.01%

by GLE protons, while the ozone changes below 50 km observed in MLS data were attributed to seasonal changes.

The MLS ozone anomalies we observe are, on contrary to the analysis in Jackman et al. 2011, not due to seasonal

changes. To identify whether the anomalies are due to direct SPE effect or not, relative ozone response from MLS

and WACCM-D simulation in MLS observation time and location to 16th Jan 2005 SPE are compared in Fig. 5.

As WACCM-D simulation are carried out in the specified dynamics mode, any dynamical variations of ozone

including ozone chemistry, are expected to be reproduced by the model well. But any direct proton impacts below

25 km would not be reproduced at all since protons with energy > 300 MeV are not included in the model input. So significant differences between the model and MLS response might indicate a direct proton impact. As shown in Fig. 5, ozone responses below 20 km are very similar between results derived from these two data sources (5 days after SPE onset, close to the GLE event), indicating no significant proton effect. We do see some differences between 20–30 km, which might demonstrate a possible direct proton effect. However, we would like to point out that compared to MLS, WACCM-D holds an > 20% overestimation of northern polar cap ozone below 30 km in January–April (see right panel of Fig.5, Fig. S4 in the supplement, and Fig. 1 in Froidevaux et al., 2019). Such differences may implicate a transport-related issue in the model (Froidevaux et al., 2019), therefore weaken our confidence to confirm the robust signal difference between MLS and WACCM-D at 20–30 km as the evidence of direct SPE impact. Readers who are interested in the ionization rate of this case is referred to Fig. A4.

Nonetheless, in our study the robust MLS ozone destruction signature in the lower stratosphere following the January 2005 SPE is unique, not only when compared to other SPEs cases after 2004, but also when large and extreme SPEs before 2004 are included (see the WACCM-D simulation result presented in Fig. 6). Further research needs to be done to confirm the dynamical/chemical factors that led to ozone destruction below 35 km in January 2005.

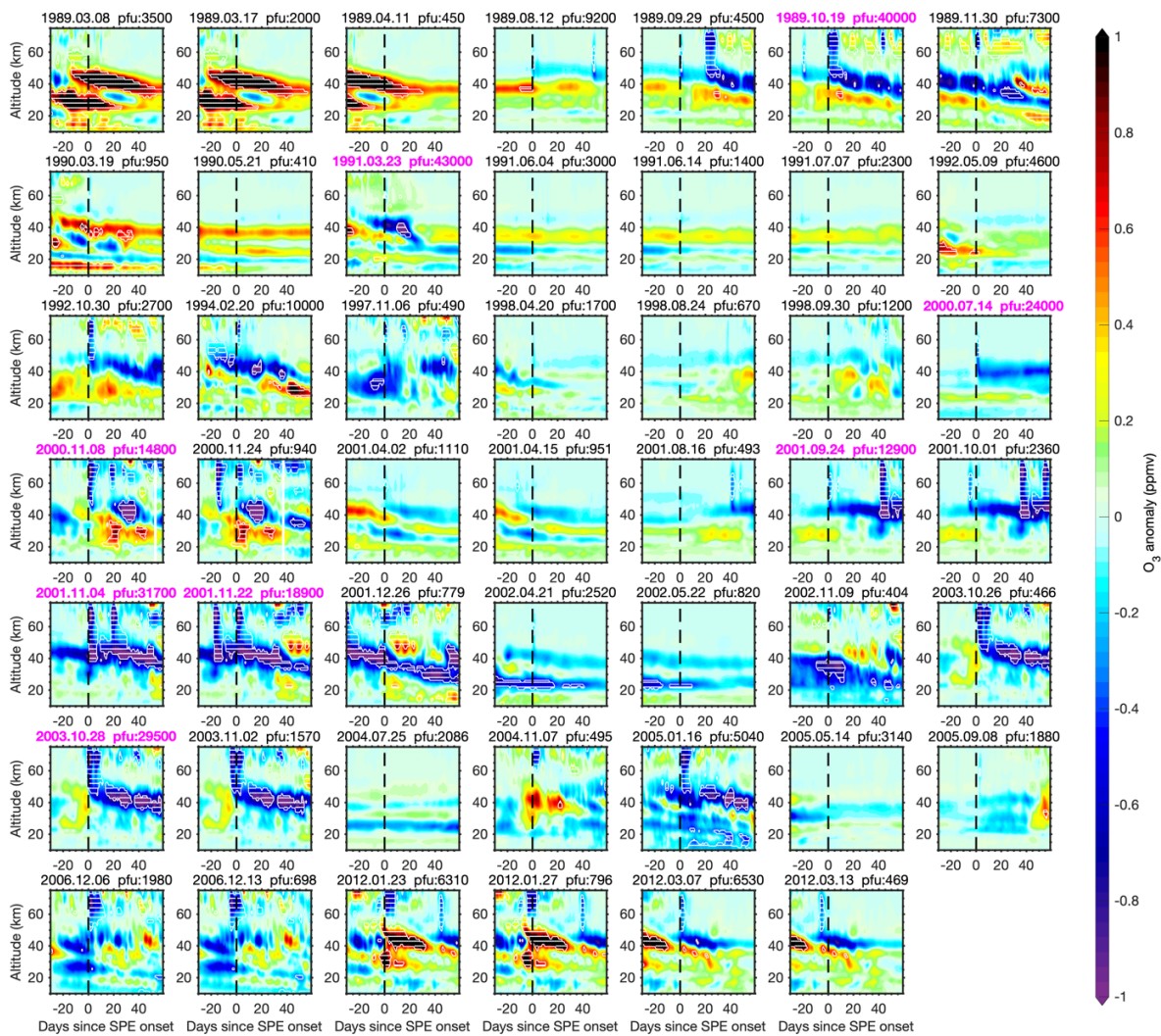

**Figure 6. Same as Fig. 4 but for all simulated WACCM-D ozone anomalies (not only collocated with MLS measurement) before and after individual big SPEs (proton fluxes >400 pfu) since 1989. Extreme SPEs (proton fluxes >10000 pfu) are marked with bold magenta titles.**

## 5    Conclusions

Recent studies have reported observations of up to 10% average decrease of lower stratospheric ozone at 20 km altitude following solar proton events (SPE). However, mechanisms which could cause such a large low-altitude impact are not clear. We used the Aura MLS satellite ozone datasets from 2004 to date and WACCM-D model simulations from 1989–2012 to analyse SPE-driven ozone changes. In our approach, stratospheric and mesospheric daily ozone anomalies (10–70 km) were examined over the epochs of SPEs by applying 1) a Superposed Epoch Analysis (SEA) for all the cases and 2) a case-by-case analysis for individual events. Statistical significance of the anomalies found in the ozone levels was estimated by employing a Monte Carlo approach.

Arctic polar ozone destruction in the mesosphere and upper stratosphere can be directly observed from satellite measurement anomaly, when following SPEs in September–April with proton fluxes >400 pfu and >1000 pfu, respectively. We observe 5–10% ozone destruction below 30 km altitude in MLS SEA results. However, the depletion appears before the epoch time, i.e. SPE onset. We argue that such lower stratospheric ozone losses are rather caused by unusually stable and strong polar vortex, together with sufficient ozone depleting reservoirs of chlorine as confirmed by the case-to-case study. In the case by case study, we find a very good overall consistency between SPE-driven ozone anomalies derived from the WACCM-D model simulations and the Aura MLS data. Despite the model can only detect direct proton effect above 25 km due to the input proton energy threshold 300 MeV, the good consistency enables us to generalise the study also to the SPEs before the Aura MLS era. From 1989 to date, robust lower stratospheric ozone decrease after SPEs was observed only once in ozone anomaly, i.e. following the January 2005 SPE. Ozone was depleted by ~1 ppmv (~15%) at 20–35 km and by ~0.15 ppmv (>20%) below 15 km 5 days after SPE onset. We further investigated this case by comparing WACCM-D and MLS data. Since WACCM-D is not expected to observe direct SPE impact below 25 km, a consistent ozone depletion below 15 km demonstrated that direct SPE impact is less likely to be the reason for this robust ozone loss. The source of ozone loss above 20 km, however, is not fully confirmed. We stress that the January 2005 event was followed by a GLE, but the SPE was not the strongest on record by far. The exact mechanisms of the suggested lower stratosphere impact following this event are currently unclear. The simulation results indicate that even for the strongest SPEs in our record, there is no significant effect on the lower stratospheric ozone as such.

Although it remains unclear to what degree the lower ozone decrease in January 2005 was caused by the SPE, and how much due to other natural variability, we suspect that the observed, statistically significant lower stratospheric ozone impact is most likely by chance coincident with the SPE onset. We note that further research on January 2005 SPE case is necessary, to solidly confirm the EPP/dynamical/chemical factors that led to ozone destruction below 35 km, but outside the scope of this paper.

1    **Appendix**

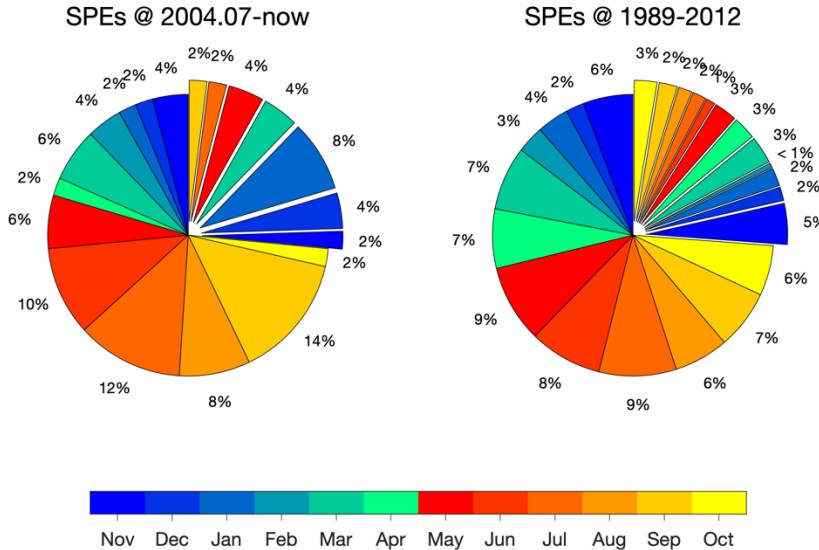

3    **Figure A1. SPE's seasonal distribution for those with fluxes >400 pfu (exploded parts of the pie chart) and the ones**
4    **with fluxes <400 pfu (regular parts of the pie chart). Left panel demonstrates the cases in between MLS measurement**
5    **period (2004.07– now). Right panel shows the cases during WACCM-D simulation (1989–2012).**

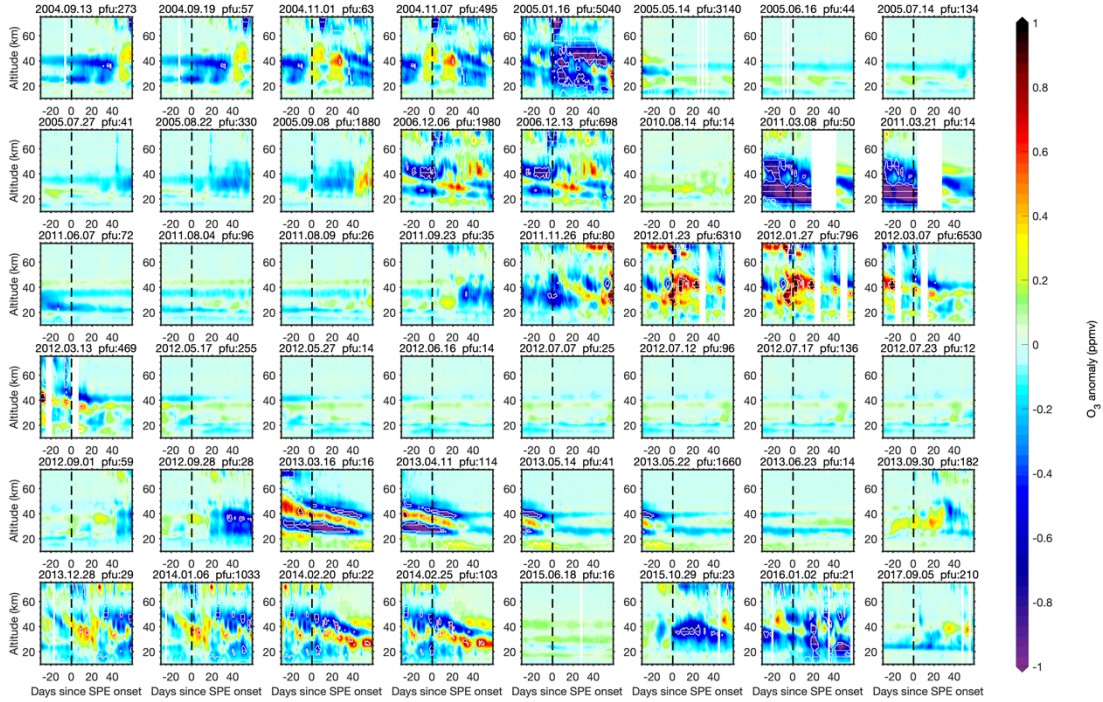

7    **Figure A2. Same as Fig. 3 but after all individual SPEs since July 2004.**

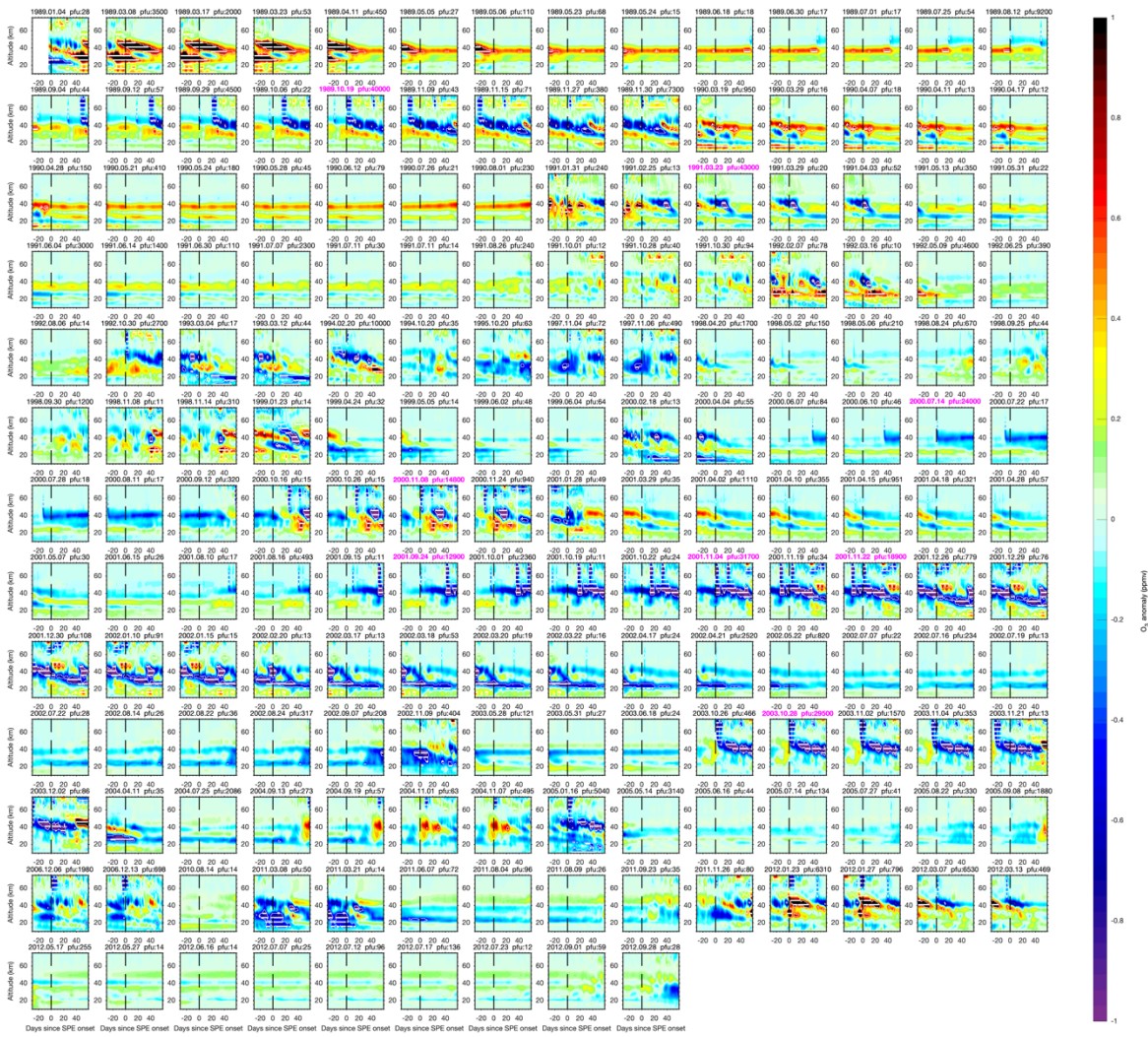

Figure A3. Same as Fig. 4 but after all individual SPEs since 1989.

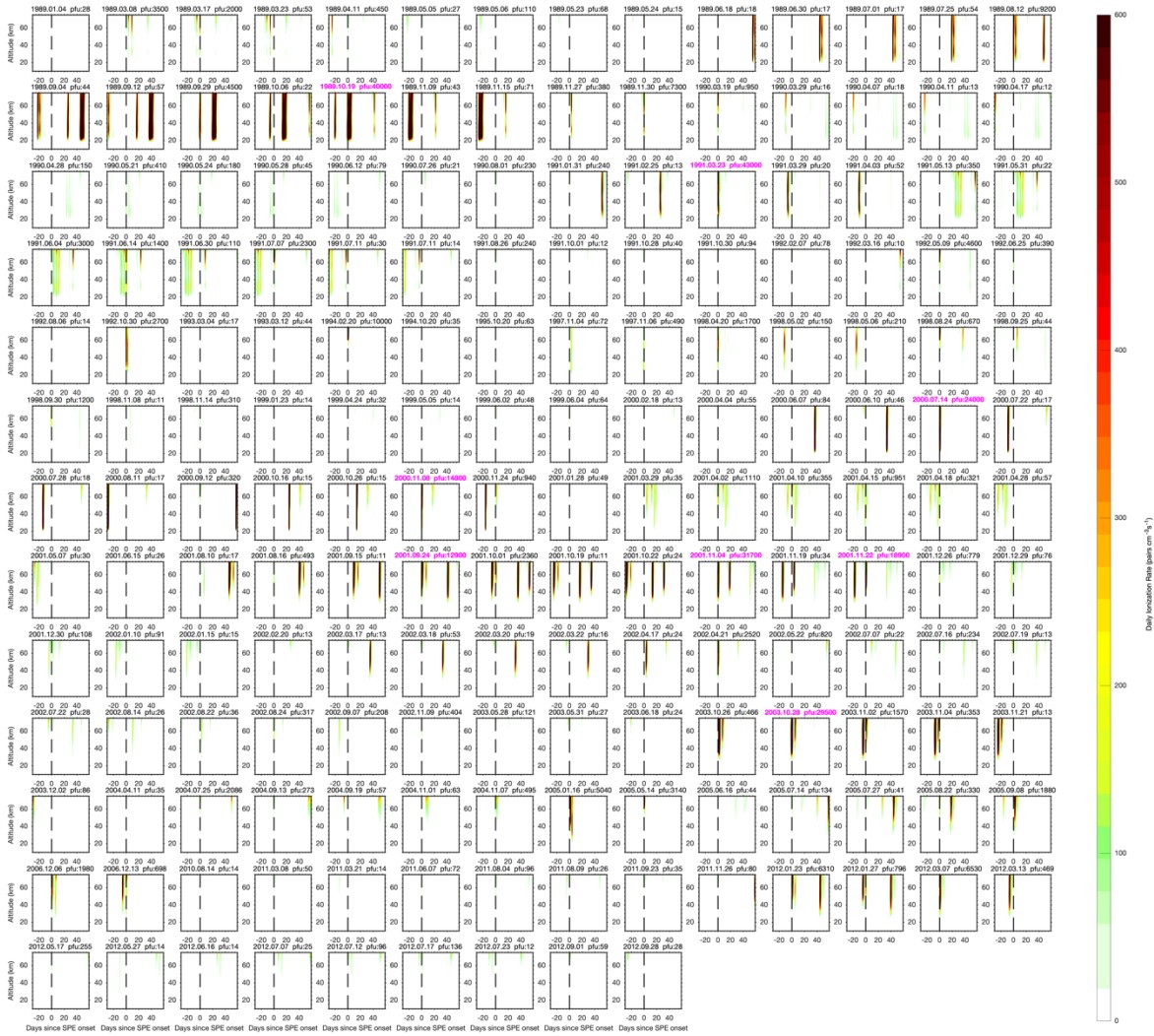

Figure A4. Daily averaged ionisation rate along with altitude at 30 days before and 60 days after individual SPEs since 1989.

**Data availability**

MLS ozone data used in this study is available at https://mls.jpl.nasa.gov/products/o3_product.php. Proton fluxes and solar proton events are available from https://www.ngdc.noaa.gov/stp/satellite/goes/index.html. Daily geomagnetic activity Ap-index used in Fig. 1 can be found at https://www.ngdc.noaa.gov/stp/GEOMAG/kp_ap.html. SPE induced ionization rate dataset is available at https://solarisheppa.geomar.de/solarprotonfluxes.

**Author contribution**

JJ and AK formed the idea of the work. JJ performed the analysis and wrote the paper with contributions from NK, PTV and AK. MES provided the WACCM-D model data. All the authors had intensive discussions about the method and results during the research.

**Competing interests**

The authors declare that they have no conflict of interest.

## Acknowledgements

J. J. is funded by the University of Oulu's Kvantum Institute. The work of A.K. is funded by the Tenure Track Project in Radio Science at Sodankylä Geophysical Observatory/University of Oulu. We would like to thank the MLS ozone teams for providing the ozone data. This work was carried out as a part of International Space Science Institute (ISSI) project "Space Weather Induced Direct Ionisation Effects On The Ozone Layer". We appreciate the fruitful discussion with the group members in this project. We would like to thank the editor and the two anonymous referees for dedicating their time to this work.

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
