# Peer review of "Is there a direct solar proton impact on lower stratospheric ozone?"

_Atmospheric Chemistry and Physics, 2020_

## Referee Comment (RC1) · Anonymous Referee #1 · 8 Jun 2020

In this paper, the impact of solar proton events on ozone in the high-latitude ($60\text{-}90^\circ$N) lower stratosphere below 30 km altitude is investigated bases on MLS observations and model simulations with the WACCM-D model. It is well known that large solar proton events can have a large impact on atmospheric composition, particularly on $NO_x$, $HO_x$ and ozone, in the upper stratosphere and mesosphere above 40 km altitude, but little evidence has been provided so far for an impact on the stratosphere below 30 km; the data shown in this study provide interesting new insight in this field. The paper is well structured and well written.

However, unfortunately in my opinion the main conclusion provided in this paper is not sufficiently supported by the results shown.

The main conclusion is that "the SPE has a zero direct impact on the lower strato-

spheric ozone" (line 20 of the abstract). This is a very strong conclusion. My opinion that it is not sufficiently supported by the results is based on three things: a) while the authors show that some of the observed anomalies in 10-20 km altitude are likely due to other forcings not SPEs, there are some cases where negative anomalies at and below 20 km occur, but no other forcings besides proton ionization are identified. This is particulary true for the ground-level event of January 2005 which had exceptionally high fluxes at high particle energies. Further analyses are clearly needed before a clear conclusion can be drawn on this. b) the lower stratosphere is affected directly only by protons with energies larger then 200 MeV, but the analysis is based on fluxes of protons with energies > 10 MeV, which are relevant in the mid- and lower mesosphere below 70 km. c) the model results are based on proton fluxes > 300 MeV which affect the atmosphere above 25 km altitude (see further comments and references below); no conclusions can be drawn from these model runs about a possible direct impact of proton ionization below 25 km altitude.

Therefore, the authors either need to carry out further analyses of their data (some suggestions are provided in my specific comments below), or formulate their conclusions much more carefully.

Specific comments:

Title and page 1, line 11: on lower stratospheric ozone depletion. I would say that "lower stratosphere" would mean tropopause (8-12 km) to about 20-25 km. However lower stratosphere is not a well-defined term, so just clarify in this sentence which altitudes you focus on (10-30 km?)

Page 2, line 12 "most advanced climate models are now including EPP forcing" better change this to "many advanced chemistry-climate models . . ." because a) I doubt that this is really done by "most" models, and b) you need atmospheric chemistry to include EPP forcing. Climate models include atmospheric dynamics and ocean coupling, but not necessarily interactive atmospheric chemistry. The term is either chemistry-climate

model or composition-climate model to clarify that you need atmospheric chemistry as well.

Page 2, line 14: please provide a reference for ∼10% alpha

Page 2, line 15 and 16: . . . tens to hundreds of MeV . . . at altitudes of 35—90 km . . .. I think what you mean is that solar proton events affect the atmosphere mainly in the altitude region of 35—90 km, but what you say is that protons and alpha of 10-1000 MeV mainly deposit their energy in 35—90 km. This is not correct. Protons with energies of 10 MeV release most of their energy around 70 km, protons with energies of 100 MeV release their energy in 30—40 km, protons with energies of 1000 MeV release their energy below 20 km (Turunen et al., 2009, Fig 3; Wissing and Kallenrode, 2009, Fig. 2). Soft protons and electrons may also contribute to affect altitudes above 70 km, and the fluxes of protons larger than 100 MeV are low in many solar proton events, though events with a very hard spectrum, with large fluxes of > 100 MeV protons, exist. One example is the very strong ground-level enhancement of January 2005 (e.g., Jackman et al., 2011, see also Table 1 in Gopalswamy et al., 2005). Please be more precise.

Page 2, Line 25 "large events", and line 27 "very extreme events", please specify what you mean by those terms. Presumably fluxes of protons at, or larger than, some specified energy range.

Page 3, line 22: 300 MeV protons mostly affect the altitude range around 25 km (Turunen et al., 2009; Wissing and Kallenrode, 2009). As 300 MeV is the upper limit of proton energies considered in your model runs, you can therefore only investigate the impact of direct proton forcing at altitudes above ∼25 km. As you don't implement proton energies able to reach altitudes below 25 km, you can not make any statements on the impact of proton ionization on altitudes below 25 km based on these model experiments. You could presumably use the model experiments to investigate possible dynamical feedbacks onto the lower stratosphere below 25 km to proton ionization

above this altitude though. Page 3, line 27-30: . . . protons > 300 MeV are not included .. as the contribution of > 300 MeV protons to direct ozone loss . . . would likely be negligible due to the small fluxes at such high energies . . . to summarize: you don't include those proton energies because they likely have no impact, do model experiments without those proton energies, analyse the model experiments, and conclude that there is no impact in those altitudes? That is circular reasoning. See my comment above: you can not draw any conclusions of a direct impact of proton ionization on altitudes below 25 km on the basis of these model experiments.

Page 4, Figure 1: the figure caption states that ionization rates used for this figure are derived at 1 hPa, that is, about 45 km – around the stratopause. If you want to investigate the impact of those events on the lower stratosphere below 30 km altitude, this is not a very useful quantity. Ionization rates at 10 hPa ($\sim$30 km) or even lower would be much more relevant here. I would suggest that you either exchange this figure with 10 hPa, or show both 10 hPa and 1 hPa.

Page 4, line 20, and page 3, line 35: the SPE onset time is defined as the time when 5-min average proton fluxes with energies > 10 MeV are greater than 10 pfu. Why base the analysis on protons of comparably low energies (10 MeV protons mainly affect $\sim$70 km altitudes, see above) if you want to analyse the impact on the lower stratosphere below 30 km? > 100 MeV would be more relevant. Even if you argue that you do not want to exclude soft-spectrum SPEs with large fluxes, the onset time of the event may vary for different energies.

Page 5, line 13-14: the spatial distribution of events is similar in summer and winter, but the amplitudes are larger during winter. This may also be a purely statistical effect due to the much lower number of events (19 compared to 49), as outliers have a larger impact in small sample sizes.

Page 5, line 24: "the signatures above and below are not related to the epoch time" what you actually see is that the signatures already appear a considerable time before

the event. So you argue that they are not related to the event. This is not necessarily true. Solar proton events are not completely isolated events. There often are series of solar proton events separated by a few (up to 27) days, as clearly seen, e.g., in Figs. 3, A2 and A3. If the first event in a series is strong, then the superposed epoch gives a response before the event. You can clearly see this in the right panel of Figure 2 at 60-70 km. Solar proton events are also often preceded by strong flares which may or may not have an impact on the atmosphere, and occur during periods of strong geomagnetic activity, with geomagnetic storms before, during or after the event. You can't exclude a significant response solely on the basis of the timing alone.

Page 6, line 3-4: However, there was no robust ozone loss below 30 km in the WACCM-D simulations; considering the limit of proton energies in the model experiments, one would not expect a direct impact of proton ionization in these model results below 25 km. However, the WACCM-D simulations could be used to analyse the observed response of MLS ozone in a more rigorous way, by doing the analysis of WACCM-D ozone with exactly the same sampling as done in MLS data – that is using the same number of events, and the same time-period for the baseline annual cycle. As WACCM-D is runs are carried out in the specified dynamics mode, any dynamical variations of ozone including ozone hole chemistry, should be reproduced by the model very well, but any direct proton impacts below 25 km would not be reproduced at all, so significant differences between model and MLS response might indicate a direct proton impact. However, if results are very similar, this would indicate no significant (on average) proton impact. This would provide a more rigorous test also than comparing the individual events in the Appendix figures, and I urge the authors to do such an analysis.

Page 6, lines 6-7: Despite the fact that WACCM-D epoch analysis . . . around 20 km . . . it is a good idea to look at individual events, but that there is no response of WACCM-D results at 20 km is the totally wrong argument here, because WACCM-D only includes proton energies > 300 MeV. A better argument would be the low number of events, and

high variability of stratospheric ozone, influenced, e.g., also by SSWs or heterogeneous chemistry on PSC surfaces, particularly during winter. Please rewrite this sentence accordingly.

Page 6, section 4 and Figures 3, 4: you select events here based on large proton fluxes > 10 MeV. However, if you really want to look at impacts on the lower stratosphere, it would make more sense to select for > 300 MeV fluxes. You could also select for ground-level events, however, based on the list provided by Gopalswamy et al 2005, this would presumably leave you with a list of 1 in the MLS time-period – January 2005, which really seems to have been exceptional (is there an update for 2005-now?).

Page 6, line 17-18: if you want to compare the variation in the MLS and WACCM-D events, it would be better to use the same period – the MLS data period – for both MLS and WACCM-D. Else differences in the anomalies might also be due to differences in the background period.

Page 7, line 6 to page 8, line 16: this analysis on the reasons for strong ozone anomalies not related to SPEs is very useful and concise. I also agree to your conclusion as stated in lines 15-16 of page 8, that these variations contribute to the robust anomalies below 30 km as seen in Fig. 2. In particular, the significant negative ozone anomaly in 10-30 km starting well before the event onset is clearly influenced strongly by the anomalously cold late winter/spring in early 2011, whose impact on lower stratospheric ozone is well documented (e.g., Sinnhuber et al., 2011). However, I think you should go one step further and redo the superposed epoch analysis excluding those events in cold winters (that is, in winter 2010/2011, 2015/2016 and 2019/2020), and also those events where an SSW occurred within the epoch period. While this would reduce the number of events, it would also reduce the background variability, and thus hopefully provide more robust results.

Page 9, lines 11-14: you should stress here that the MLS anomalies you observe are (contrary to the analysis in Jackman et al 2011) not due to seasonal changes. The

changes you observe during and after the January 2005 GLE may be unique within the MLS timeseries; however, so apparently is the event itself, at least in terms of the highest energies (compare to Tab 1 in Gopalswamy et al 2005). It may be comparable in terms of the > 10 MeV fluxes compared to the Oct-Nov 2003 SPEs, but in terms of the highest energies, fluxes were apparently much larger – more than an order of magnitude in terms of the GLE intensity. So the ozone changes observed during this event below 20 km altitude might indicate that ozone losses related to SPEs in these altitudes may be possible for events with very hard spectra (GLEs). I agree with you that more research needs to be done on this before a robust conclusion can be drawn on this, but I don't think you can dismiss this on the basis that no other event shows something similar. It appears to be a fairly unique event.

Page 9, lines 36-39: I do not agree that you can draw the conclusion that "bases on our analysis . . . SPE do not cause direct lower stratospheric ozone anomalies" based on the evidence you have provided. I agree that you provided evidence that some of the significant negative ozone anomalies are not due to a direct ozone impact but to other forcings, most obviously in March 2011. However, you do not provide a similar convincing explanation for the January 2005 event, which was exceptional in containing a very hard spectrum, and thus provides the most likely candidate of an impact on the lower stratosphere from the events sampled here. This of course does not prove that such an impact exists for very hard spectra ground-level events in general, or even during this event. However, you can't just disregard it, either; there clearly is a need for further analysis on this topic. You can conclude that solar proton events with large fluxes at > 10 MeV do not necessarily provide a large impact below 30 km altitude, if they don't have a very hard spectrum with high fluxes at > 200 MeV as well. However, you can't say anything definite about hard-spectra solar proton events here because you did not explicitly test for this.

Turunen et al., Impact of different energies of precipitating particles on NOx generation in the middle and upper atmosphere during geomagnetic storms, JASTP, 71, 1176-

Interactive
comment

1189, 2009

Wissing, J.M. and Kallenrode, M., Atmospheric Ionization Module Osnabrueck (AIMOS), J. Geophys. Res., 114, doi: 10.1029/2008JA013884, 2009.

Jackman, C.H., et al., Northern hemisphere atmospheric influence of solar proton events and ground level enhancement in January 2005, Atmos. Chem. Phys., 11, 6153-6166, 2011

Gopalswamy, N., et a., Coronal mass ejections and ground level enhancements, 29th Cormic Ray Conference Pune, India, 1, 101-104, August 2005

Sinnhuber, B.-M., et al., Arctic winter 2010/2011 at the brink of an ozone hole, GRL, 38, https://doi.org/10.1029/2011GL049784, 2011

---

## Referee Comment (RC2) · Anonymous Referee #2 · 9 Jun 2020

The manuscript is a response to two previous studies by Denton et al. (2018a,b) which reported up to 10% average decrease of ozone at ~20 km following solar proton events. Applying the same method (superposed epoch analysis) on different ozone observations (MLS, Aura), the current study arrive at a different conclusion: "SPE do not cause direct lower stratospheric ozone anomalies", which is corroborated by both observed and modeled case studies. The paper is well written and logically organized. Nevertheless, the manuscript still holds the potential for improvement, both in regard to the methods applied and the subsequent discussion.

Major revisions

1) Selection of events for the superposed epoch analysis:

[Figure]

The superposed epoch analysis and case studies suffer from the lack of isolated SPEs. Several years have multiple SPEs occurring days apart. That implies that the period before the zero epoch time is already influenced by SPEs. Despite lower statistics, it would be more accurate to select one event, possible the first, within "the time frame". The same argument applies to the case studies, where one period could be marked with several onsets to avoid reproduction of the "same figure". Further, Figure 2 and the case studies would be more informative if the estimated ionization rates were added.

2) Ozone anomalies in respect to climatology:

Ozone anomalies are evaluated in respect to the climatology. The case studies demonstrates, as pointed out in the discussion, that the year to year dynamical variability is larger than the potential ozone impact. These conditions make it impossible to conclude that SPEs has zero impact on ozone. It is only possible to conclude that it is less than the year to year dynamical variability. It also demonstrates that the climatology is not necessarily a good reference frame to evaluate the SPE-impact. E.g. one of the strongest SPE, with onset 2012.03.07, has a strong positive ozone anomaly before the event which becomes less positive after the event. Hence it might be a reduction compared to the pre-storm values. Also, in Figure 2 (the superposed epoch analysis) single years such as January 2012 is evident as a significant positive anomaly below 40 km. Hence, I speculate if the SPE impact would be better represented as a change relative to the ∼20 days preceding the event. (Alternatively, events dominated by extreme dynamical anomalies such as January 2012 should be excluded from the superposed epoch analysis.)

3) Proton energy range in the WACCM model:

For the model runs, only protons with <300 MeV are included in the ionization rates. The respective energy range is therefore insufficient to account for the direct impact at ∼20 km (e.g. Turunen et al., 2009). Without a complete energy range impacting 20 km, the discussion and the subsequent conclusion should reflect this limitation. It should

also be noted that the 2005.01.16 case study are more pronounced in the observations compared to the model. This is particular true below 30 km, which might imply that the model might underestimate the ionization rates, transport or chemical processes.

4) Ozone chemistry:

Would you expect the same chemistry to impact ∼20 km altitude as ∼70 km? Is it still only EPP produced NOx and HOx than deplete ozone as described in the introduction, or are the chemical pathways of more complex deep into the lower stratosphere? E.g. Jackman et al. (2000) suggest that enhanced NOx values can lead to enhanced formation of the chlorine and bromine reservoir species ClONO2 and BrONO2, slowing down the 'ozone hole' formation chemistry in cold polar winters.

Minor revisions:

1 Introduction: Define altitude range of upper and lower stratosphere Line 12: define altitude range of lower stratosphere Line 13: Define acronym when "superposed epoch analysis" are first written (Line 12) Introduction: define altitude range of upper and lower stratosphere Line 18: "at many altitudes" be more precise Line 35-37: Outline where the results from WACCM is coming

2 Data sets Line 3: remove Microwave Limb Sounder as acronym is already defined Line 1/7 page 4: add (∼50 km) the first time you write ∼1 hPa

References, page 14, line 4: remove hyphen

---

## Author Comment (AC1) · 6 Aug 2020

Answers to comments by anonymous referees to manuscript acp-2020-273: Jia et al., Is there a direct solar proton impact on lower stratospheric ozone?

We would like to thank all the anonymous referees for dedicating their time to the comments and the discussion. We have made our conclusions much more carefully accordingly. Some further analysis and discussion were added as well. Please find our answers and responses to the comments below.

Anonymous referee #1

Specific comments: Title and page 1, line 11: on lower stratospheric ozone depletion. I would say that "lower stratosphere" would mean tropopause (8-12 km) to about 20-

25 km. However lower stratosphere is not a well-defined term, so just clarify in this sentence which altitudes you focus on (10-30 km?)

The lower stratospheric ozone is clarified as altitudes 10-30 km now.

Page 2, line 12 "most advanced climate models are now including EPP forcing" better change this to "many advanced chemistry-climate models :::" because a) I doubt that this is really done by "most" models, and b) you need atmospheric chemistry to include EPP forcing. Climate models include atmospheric dynamics and ocean coupling, but not necessarily interactive atmospheric chemistry. The term is either chemistry-climate model or composition-climate model to clarify that you need atmospheric chemistry as well.

Modified. Thank you.

Page 2, line 14: please provide a reference for 10% alpha

The specific number of alpha in SPE can be found, for example, in the book 'Health Physics in the 21st Century' published in 2008, by Joseph John Bevelacqua. This number is irrelevant to the paper, we have deleted this information instead of adding a reference.

Page 2, line 15 and 16: . . . tens to hundreds of MeV . . . at altitudes of 35-90 km... I think what you mean is that solar proton events affect the atmosphere mainly in the altitude region of 35-90 km, but what you say is that protons and alpha of 10-1000 MeV mainly deposit their energy in 35-90 km. This is not correct. Protons with energies of 10 MeV release most of their energy around 70 km, protons with energies of 100 MeV release their energy in 30-40 km, protons with energies of 1000 MeV release their energy below 20 km (Turunen et al., 2009, Fig 3; Wissing and Kallenrode, 2009, Fig. 2). Soft protons and electrons may also contribute to affect altitudes above 70 km, and the fluxes of protons larger than 100 MeV are low in many solar proton events, though events with a very hard spectrum, with large fluxes of > 100 MeV protons, exist. One

example is the very strong ground-level enhancement of January 2005 (e.g., Jackman et al., 2011, see also Table 1 in Gopalswamy et al., 2005). Please be more precise.

The sentence has been modified to 'Such high-energy particles mainly affect the atmosphere at altitudes of 35-90 km, ...'

Page 2, Line 25 "large events", and line 27 "very extreme events", please specify what you mean by those terms. Presumably fluxes of protons at, or larger than, some specified energy range.

The terms are specified now.

Page 3, line 22: 300 MeV protons mostly affect the altitude range around 25 km (Turunen et al., 2009; Wissing and Kallenrode, 2009). As 300 MeV is the upper limit of proton energies considered in your model runs, you can therefore only investigate the impact of direct proton forcing at altitudes above 25 km. As you don't implement proton energies able to reach altitudes below 25 km, you can not make any statements on the impact of proton ionization on altitudes below 25 km based on these model experiments. You could presumably use the model experiments to investigate possible dynamical feedbacks onto the lower stratosphere below 25 km to proton ionization above this altitude though. Page 3, line 27-30: ... protons > 300 MeV are not included .. as the contribution of > 300 MeV protons to direct ozone loss . . . would likely be negligible due to the small fluxes at such high energies . . . to summarize: you don't include those proton energies because they likely have no impact, do model experiments without those proton energies, analyze the model experiments, and conclude that there is no impact in those altitudes? That is circular reasoning. See my comment above: you can not draw any conclusions of a direct impact of proton ionization on altitudes below 25 km on the basis of these model experiments.

Agreed. The model certainly cannot detect proton caused changes below 25 km without sufficient particle input. We have stressed the statement more carefully in Sect. 2.2, so that the readers are aware of this limitation: The sentence "We also stress that

protons >300 MeV are not included in the simulation, as the contribution of >300 MeV protons to direct ozone loss in the lower stratosphere would likely be negligible due to the relatively small fluxes at such high energies (Jackman et al., 2011)" is modified to "We also stress that protons >300 MeV are not included in the simulation. 300 MeV protons mostly affect the atmosphere at around 25 km (Turunen et al., 2009; Wissing and Kallenrode, 2009). As 300 MeV is the upper limit of proton energies considered in our model simulation, the WACCM-D simulation presented here can therefore only investigate the impact of direct proton forcing at altitudes above âĹij25 km".

Page 4, Figure 1: the figure caption states that ionization rates used for this figure are derived at 1 hPa, that is, about 45 km – around the stratopause. If you want to investigate the impact of those events on the lower stratosphere below 30 km altitude, this is not a very useful quantity. Ionization rates at 10 hPa (âĹij30 km) or even lower would be much more relevant here. I would suggest that you either exchange this figure with 10 hPa, or show both 10 hPa and 1 hPa.

Agreed. We have kept the original figure as the upper panel and added the ionization rate at ∼12 hPa (pressure level in ionization rate data) as the lower panel in Figure 1. The annotation is adjusted accordingly.

Page 4, line 20, and page 3, line 35: the SPE onset time is defined as the time when 5-min average proton fluxes with energies > 10 MeV are greater than 10 pfu. Why base the analysis on protons of comparably low energies (10 MeV protons mainly affect âĹij70 km altitudes, see above) if you want to analyze the impact on the lower stratosphere below 30 km? > 100 MeV would be more relevant. Even if you argue that you do not want to exclude soft-spectrum SPEs with large fluxes, the onset time of the event may vary for different energies.

We agree that using > 100 MeV proton fluxes could be more accurate and relevant for our study. The SPE onset was chosen in this way to keep consistency with Denton et al. The results from individual SPE studies also showed that our conclusion will not be

influenced by changing the definition of SPE onset. The figure below shows the GOES pfu of >10 MeV protons (black line) and > 100 MeV protons (blue line) respectively, while the red line represents our current SPE onsets. The timing of the events does not change significantly. We argue that defining SPE onset using 10 MeV protons or 100 MeV protons is not very critical.

Support figure: GOES proton fluxes example with energy threshold of 10 MeV and 100 MeV in 2000-2005.

Page 5, line 13-14: the spatial distribution of events is similar in summer and winter, but the amplitudes are larger during winter. This may also be a purely statistical effect due to the much lower number of events (19 compared to 49), as outliers have a larger impact in small sample sizes.

Indeed. We address the statistical effect in the manuscript now.

Page 5, line 24: "the signatures above and below are not related to the epoch time" what you actually see is that the signatures already appear a considerable time before the event. So you argue that they are not related to the event. This is not necessarily true. Solar proton events are not completely isolated events. There often are series of solar proton events separated by a few (up to 27) days, as clearly seen, e.g., in Figs. 3, A2 and A3. If the first event in a series is strong, then the superposed epoch gives a response before the event. You can clearly see this in the right panel of Figure 2 at 60-70 km. Solar proton events are also often preceded by strong flares which may or may not have an impact on the atmosphere, and occur during periods of strong geomagnetic activity, with geomagnetic storms before, during, or after the event. You can't exclude a significant response solely on the basis of the timing alone.

Agreed. As is suggested by reviewer 2, we have now re-calculated the statistical response by keeping the first event only, when several events happened within 10 days. This will partly exclude the possible response before epoch time from previous SPEs in a nearby time frame. Nevertheless, "the signatures above and below are not related

[Figure]

to the epoch time" is removed from the manuscript.

Page 6, line 3-4: However, there was no robust ozone loss below 30 km in the WACCM-D simulations; considering the limit of proton energies in the model experiments, one would not expect a direct impact of proton ionization in these model results below 25 km. However, the WACCM-D simulations could be used to analyze the observed response of MLS ozone in a more rigorous way, by doing the analysis of WACCM-D ozone with exactly the same sampling as done in MLS data – that is using the same number of events, and the same time period for the baseline annual cycle. As WACCM-D runs are carried out in the specified dynamics mode, any dynamical variations of ozone including ozone hole chemistry, should be reproduced by the model very well, but any direct proton impacts below 25 km would not be reproduced at all, so significant differences between model and MLS response might indicate a direct proton impact. However, if results are very similar, this would indicate no significant (on average) proton impact. This would provide a more rigorous test also than comparing the individual events in the Appendix figures, and I urge the authors to do such an analysis.

This is a great idea. When re-analyzing ozone's responses to SPE in Figs 3 and 4, WACCM-D profiles were output at Aura Microwave Limb Sounder (MLS) observation times and locations. Climatology from MLS and WACCM-D are calculated from the same time period (2nd August 2004 – 31th December 2012). We provide a comparison result between MLS and WACCM-D for the individual case study in January 2005 as the new Fig.5 in the manuscript, the discussion is added accordingly.

We keep analyzing statistical response result from MLS using all possible SPEs for statistical reasons in the manuscript. A similar epoch analysis using MLS and WACCM-D simulation at MLS observation time and location during their overlapping period, as is suggested by the reviewer, is added to the supplement as Fig. S1.

We would like to point out an inconsistency between ozone from MLS and the used model below 30 km. We added a comparison of WACCM-D simulation at MLS time

and location, and MLS daily ozone anomalies in the polar cap in the supplement (Fig. S4). WACCM-D model overestimates northern polar cap ozone by 10% to > 20% below 30 km in January-April. This is consistent with SD-WACCM ozone vs MLS ozone results reported in Froidevaux et al., 2019. Such difference may implicate a transport-related issue in the model (Froidevaux et al., 2019), thus, weaken our confidence of the reviewer's idea that 'significant differences between model and MLS response might indicate a direct proton impact'.

Page 6, lines 6-7: Despite the fact that WACCM-D epoch analysis . . . around 20 km . . . it is a good idea to look at individual events, but that there is no response of WACCM-D results at 20 km is the totally wrong argument here, because WACCM-D only includes proton energies > 300 MeV. A better argument would be the low number of events, and high variability of stratospheric ozone, influenced, e.g., also by SSWs or heterogeneous chemistry on PSC surfaces, particularly during winter. Please rewrite this sentence accordingly.

Amended. We appreciate the suggestion.

Page 6, section 4 and Figures 3, 4: you select events here based on large proton fluxes > 10 MeV. However, if you really want to look at impacts on the lower stratosphere, it would make more sense to select for > 300 MeV fluxes. You could also select for ground-level events, however, based on the list provided by Gopalswamy et al 2005, this would presumably leave you with a list of 1 in the MLS time-period – January 2005, which really seems to have been exceptional (is there an update for 2005-now?).

We have ground-level data till April 2017. In the MLS time period, there is no event that is comparable with the January 2005 event. There is a smaller event in December 2012, right in between two SPE events that I used. We didn't see visible ozone abnormal that are at a 95% confidential level. January 2005 is quite exceptional and is worth to be checked more carefully in the future.

Page 6, line 17-18: if you want to compare the variation in the MLS and WACCM-D

events, it would be better to use the same period – the MLS data period – for both MLS and WACCM-D. Else differences in the anomalies might also be due to differences in the background period.

We agree. The results in Fig. 3 (from MLS) and Fig. 4 (from WACCM-D) are adjusted by using data at the overlapping period, i.e., 2nd August 2004 – 31th December 2012 (Currently our WACCM-D results are till the end of 2012 only). Moreover, to make it more comparable, the WACCM-D profiles used in Fig. 4 are also changed to the WACCM-D profile outputs at MLS observation time and location, as mentioned in the response to comment on Page 6 line 3-4. Figs. 5 (i.e., Fig. 6 after paper modification), A2 and A3 are kept the same, that is to say, these results are calculated as described in the discussion paper Page 6, line 17-18. Result description in Sect. 4 are revised accordingly.

Page 7, line 6 to page 8, line 16: this analysis on the reasons for strong ozone anomalies not related to SPEs is very useful and concise. I also agree to your conclusion as stated in lines 15-16 of page 8, that these variations contribute to the robust anomalies below 30 km as seen in Fig. 2. In particular, the significant negative ozone anomaly in 10-30 km starting well before the event onset is clearly influenced strongly by the anomalously cold late winter/spring in early 2011, whose impact on lower stratospheric ozone is well documented (e.g., Sinnhuber et al., 2011). However, I think you should go one step further and redo the superposed epoch analysis excluding those events in cold winters (that is, in winter 2010/2011, 2015/2016 and 2019/2020), and also those events where an SSW occurred within the epoch period. While this would reduce the number of events, it would also reduce the background variability, and thus hopefully provide more robust results.

This is a good suggestion. We agree that the background variability will be reduced if days with unstable polar conditions are excluded (e.g., cold winters and SSWs). However, we have to consider that for a statistical ozone study, removing data using such selecting method will introduce bias to the background variation. For instance, if

the removed data are not balanced from eastly/westly QBO years, bias from the QBO signal will be brought in. To avoid that, we need to discuss eastly/westly QBO years separately. With the current amount of SPE events we have, it is not necessary to go to such complicated selecting criteria yet.

Page 9, lines 11-14: you should stress here that the MLS anomalies you observe are (contrary to the analysis in Jackman et al 2011) not due to seasonal changes. The changes you observe during and after the January 2005 GLE may be unique within the MLS time series; however, so apparently is the event itself, at least in terms of the highest energies (compare to Tab 1 in Gopalswamy et al 2005). It may be comparable in terms of the > 10 MeV fluxes compared to the Oct-Nov 2003 SPEs, but in terms of the highest energies, fluxes were apparently much larger – more than an order of magnitude in terms of the GLE intensity. So the ozone changes observed during this event below 20 km altitude might indicate that ozone losses related to SPEs in these altitudes may be possible for events with very hard spectra (GLEs). I agree with you that more research needs to be done on this before a robust conclusion can be drawn on this, but I don't think you can dismiss this on the basis that no other event shows something similar. It appears to be a fairly unique event.

Thank you for the comment. We have now stressed the contravention with Jackman et al., 2011 regarding the January 2005 event. The conclusion is modified as well to note that there is a possibility of lower stratospheric ozone's response to protons with the highest energies.

Page 9, lines 36-39: I do not agree that you can draw the conclusion that "bases on our analysis . . . SPE do not cause direct lower stratospheric ozone anomalies" based on the evidence you have provided. I agree that you provided evidence that some of the significant negative ozone anomalies are not due to a direct ozone impact but to other forcings, most obviously in March 2011. However, you do not provide a similar convincing explanation for the January 2005 event, which was exceptional in containing a very hard spectrum, and thus provides the most likely candidate of an impact on the

lower stratosphere from the events sampled here. This of course does not prove that such an impact exists for very hard spectra ground-level events in general, or even during this event. However, you can't just disregard it, either; there clearly is a need for further analysis on this topic. You can conclude that solar proton events with large fluxes at > 10 MeV do not necessarily provide a large impact below 30 km altitude, if they don't have a very hard spectrum with high fluxes at > 200 MeV as well. However, you can't say anything definite about hard-spectra solar proton events here because you did not explicitly test for this.

Agreed. The conclusion is modified accordingly.

Anonymous referee #2

The manuscript is a response to two previous studies by Denton et al. (2018a,b) which reported up to 10% average decrease of ozone at âĹij20 km following solar proton events. Applying the same method (superposed epoch analysis) on different ozone observations (MLS, Aura), the current study arrives at a different conclusion: "SPE do not cause direct lower stratospheric ozone anomalies", which is corroborated by both observed and modeled case studies. The paper is well written and logically organized. Nevertheless, the manuscript still holds the potential for improvement, both in regard to the methods applied and the subsequent discussion.

Major revisions 1) Selection of events for the superposed epoch analysis: The superposed epoch analysis and case studies suffer from the lack of isolated SPEs. Several years have multiple SPEs occurring days apart. That implies that the period before the zero epoch time is already influenced by SPEs. Despite lower statistics, it would be more accurate to select one event, possible the first, within "the time frame". The same argument applies to the case studies, where one period could be marked with several onsets to avoid reproduction of the "same figure". Further, Figure 2 and the case studies would be more informative if the estimated ionization rates were added.

Agreed. We have now re-calculated the statistical response by keeping the first event

when several events happened within a 'time frame' of 10 days. With this limitation, the selected SPE events went down from 49 to 35.

We would like to keep the case-study figures as they are. We think the figures are capable to explain the close-by SPE onset themselves. Although there is a reproduction of almost the 'same figure', the event onset is rather clear than being squeezed into one figure. It allows us to better mark particle fluxes information as well.

We agree that the ionization rates will be useful information. The ionization rates are added to Fig 2. We are able to provide the ionization rates for WACCM-D case studies in the appendix as Fig. A4. We also provided a statistical response comparison between MLS and WACCM-D during their overlapping time in the supplement, the corresponding ionization rate average is added as the right panel of new Fig. S1.

2) Ozone anomalies in respect to climatology: Ozone anomalies are evaluated in respect to the climatology. The case studies demonstrates, as pointed out in the discussion, that the year to year dynamical variability is larger than the potential ozone impact. These conditions make it impossible to conclude that SPEs has zero impact on ozone. It is only possible to conclude that it is less than the year to year dynamical variability. It also demonstrates that the climatology is not necessarily a good reference frame to evaluate the SPE-impact. E.g. one of the strongest SPE, with onset 2012.03.07, has a strong positive ozone anomaly before the event which becomes less positive after the event. Hence it might be a reduction compared to the pre-storm values. Also, in Figure 2 (the superposed epoch analysis) single years such as January 2012 is evident as a significant positive anomaly below 40 km. Hence, I speculate if the SPE impact would be better represented as a change relative to the âĹij20 days preceding the event. (Alternatively, events dominated by extreme dynamical anomalies such as January 2012 should be excluded from the superposed epoch analysis.)

Using ∼20 days before the event as the background instead of the climatology is an interesting point. We agree this is probably a better option if the epoch frame is a

rather short period. However, for an epoch time frame of 90 days, it is quite dangerous to be used as the background to detect SPE impact, since the ozone annual variation (especially in the stratosphere) is very large. We think climatology is the right choice for a three months epoch time frame. Other dynamical variations are larger than the potential SPE impact is a fact that cannot be avoided. Using ~20 days before the event as background could potentially induce an artificial signal caused by the natural decay of the dynamical variations as well. For the alternative suggestion, please see our response to reviewer 1's comment on Page 7, line 6 to page 8, line 16.

3) Proton energy range in the WACCM model: For the model runs, only protons with <300 MeV are included in the ionization rates. The respective energy range is therefore insufficient to account for the direct impact at âĹij20 km (e.g. Turunen et al., 2009). Without a complete energy range impacting 20 km, the discussion and the subsequent conclusion should reflect this limitation. It should also be noted that the 2005.01.16 case study are more pronounced in the observations compared to the model. This is particular true below 30 km, which might imply that the model might underestimate the ionization rates, transport or chemical processes.

Agreed. We now stress the limitation caused by proton < 300 MeV in Sect. 2.2, discussion and the conclusion section.

By accepting both reviewer 1 and 2's suggestion, we added a comparison of WACCM-D (at MLS measurement time and location) and MLS ozone anomalies in the polar cap in the supplement (Fig. S4). We point out that WACCM-D model overestimates northern polar cap ozone by 10% to > 20% below 30 km in January-April. This is consistent with SD-WACCM ozone vs MLS ozone results reported in Froidevaux et al., 2019. Such difference may implicate a transport-related issue in the model (Froidevaux et al., 2019).

4) Ozone chemistry: Would you expect the same chemistry to impact âĹij20 km altitude as âĹij70 km? Is it still only EPP produced NOx and HOx than deplete ozone as

described in the introduction, or are the chemical pathways of more complex deep into the lower stratosphere? E.g. Jackman et al. (2000) suggest that enhanced NOx values can lead to enhanced formation of the chlorine and bromine reservoir species ClONO2 and BrONO2, slowing down the 'ozone hole' formation chemistry in cold polar winters.

Yes, the chemistry is different in the lower stratosphere. We do expect an increased O3 by increasing NOx at this altitude. While we keep open to any kind of ozone changes (either increase or decrease) related to SPEs, we have added the statement into the manuscript in paragraph 3 of the introduction. We appreciate the comment.

Minor revisions: 1 Introduction: Define altitude range of upper and lower stratosphere Line 12: define altitude range of lower stratosphere Line 13: Define acronym when "superposed epoch analysis" are first written (Line 12) Introduction: define altitude range of upper and lower stratosphere Line 18: "at many altitudes" be more precise Line 35-37: Outline where the results from WACCM is coming 2 Data sets Line 3: remove Microwave Limb Sounder as acronym is already defined Line 1/7 page 4: add (âĹij50 km) the first time you write âĹij1 hPa References, page 14, line 4: remove hyphen

Modified accordingly. Thank you.

Reference: Froidevaux, L., Kinnison, D. E., Wang, R., Anderson, J., and Fuller, R. A.: Evaluation of CESM1 (WACCM) free-running and specified dynamics atmospheric composition simulations using global multi-species satellite data records, Atmos. Chem. Phys., 19, 4783–4821, https://doi.org/10.5194/acp-19-4783-2019, 2019.

[Figure]

[Figure]

**Fig. 1.** Support figure: GOES proton fluxes example with energy threshold of 10 MeV and 100 MeV in 2000-2005.

---

## Author Response (AR2)

**Point-to-point answer to the comments of the revised submission to manuscript acp-2020-273: Jia et al., Is there a direct solar proton impact on lower stratospheric ozone?**

**Anonymous referee #1**

*I am well content with the changes made, which addressed all the points I raised in the first round, and which (I think) improved the paper a great deal; it is now ready for publication. Below are listed a few technical points I found.*

*Abstract, lines 20-21: "The simulation results before the Aura MLS era indicate no significant effect on the lower stratospheric ozone either" – while this is factually correct, and you mention that you can't expect this due to the energy limitation, the way it is put here gives too much weight to this sentence. Maybe you could turn around the last three sentences – it still would state the same things, but would sound slightly different: "Due to the input proton energy threshold of > 300 MeV, the model can only detect direct proton effects above 25 km, and simulation results before the Aura MLS era indicate no significant effect on the lower stratospheric ozone. However, we find a very good overall consistency between model results and MLS observations of SPE-driven ozone anomalies both on average, and for the individual cases including January 2005." ... one might argue, by the way, that this means the ozone deficit in the lowermost stratosphere after the January 2005 event is more likely due to a dynamical feedback. I'll leave it to you whether you want to add that.*

According to the suggestion, the sentence is modified to 'Due to the input proton energy threshold of > 300 MeV, the WACCM-D model can only detect direct proton effects above 25 km, and simulation results before the Aura MLS era indicate no significant effect on the lower stratospheric ozone. However, we find a very good overall consistency between WACCM-D simulations and MLS observations of SPE-driven ozone anomalies, both on average and for the individual cases including January 2005.'

*Page 3, line 9: more likely the size of the satellite footprint, not the resolution. At least across track; the distance between two footprints in the longitudinal will be much larger than 500 km.*

We re-address MLS as 'Vertical profiles are retrieved from the MLS observations with a 165 km horizontal spacing at altitudes between 8 and 90 km, a spatial resolution of ~400 km horizontal and ~3.2 km vertical'.

*Page 4, line 10: before the next event happens – you mean before the next event starts.?*

Yes, modified to 'before the next event starts'.

*Page 4, line 11: What do you mean by "The ionization rates to the atmosphere"? Do you mean "The atmospheric ionization rates were than derived by averaging ... "? Please clarify.*

The sentence is changed to 'The average ionization rates in Fig. 1 were then derived by averaging the ionization rates at 1 hPa / 12 hPa during this period'.

*Page 4, line 13: ... 177 events that occurred in "the complete" WACCM-D simulation period.*
*Page 5, line 3: "compositing" – you mean "composite analysis" I think. At least that's the other name I know this by.*
*Page 5, line 13: Note the formatting of the reference (Denton et al.), the year is missing.*

Modified accordingly. Thank you.

*Page 5, Figure 15: Note you stated that the threshold for SPEs is 2 cm-1s-1, but in the figure the threshold of visibility appears to be much higher, probably 20 cm-1s-1? Can you adapt that, please?*

We are aware of the differences. Note that the threshold used in the SPE section are specified (not necessary the exact value) to define an ending point of the SPE (it was not used to define the starting point of the SPE), so the duration and its average impact to the atmosphere as 'average ionization rate' can be estimated. The consistence of this threshold to the following figures is unnecessary.

Nevertheless, we show here the same SEA of the ionization rates using a different color bar. The upper panels show the ionization rate values that are larger than 0 cm-1s-1, while the lower panels show the ones that are larger than 2 cm-1s-1.

We would like to maintain the plot with >20 cm-1s-1 visibility in the paper draft.

[Figure]

**Support figure: Same as the lower panels of Fig. 2 in the draft but presented with a different color bar.**

*Page 6, line 1: "that these extracted signatures are likely related to SPE" …. "that these extracted signatures are likely not random, but driven by some external forcing." Note that particularly during polar winter, this might be dynamics/temperature as well.*

*Page 6, line 6: "occurred" – "occurring"*

Modified. Thank you.

*Page 6, lines 18-20: "However, since this variation starts already several days before the epoch, we cannot exclude the possibility …" as you now have restricted yourself to isolated SPEs, you can make this statement.*

Agreed and modified.

*Page 7, line 18: true, but also true for the superposed epoch; significant results mean that they are probably not random, but that does not necessarily mean they are driven by the SPE.*

Agreed.

*Page 9, Figure 5: interesting Figure. What strikes me here: a) WACCM and MLS are really remarkably similar, at least qualitatively. B) The ozone deficit below ~20 km in WACCM can not be due to a direct impact of the proton forcing, suggesting some dynamical or radiative (how, though?) feedback. C) all*

*ozone deficits below 40 km occur a few (~5?) days after the event onset – is that the day of the GLE? Or does that suggest a dynamical effect?*

They are remarkably similar. I have done the comparison for all the overlapping events, they agreed in a great level. – *is that the day of the GLE?* Yes, it is, which is exciting, but we cannot excluded the possibility of coincident. Future study needs to be done.

*Page 10, line 14/15: yes I agree – that is an interesting event.*
*Page 11, line 6: Recent studies have reported "observations of" up to 10% ....*
*Page 11, line 16-18: ... as confirmed by the case-by-case study.*
*Page 11, line 26: "over 20 km" – "above 20 km", actually, what you mean is "below 25 km", as there is an ozone deficit after the event down to 15 km, and this can't be explained by direct proton forcing (in WACCM) either.*
*Page 11, lines 28-29: maybe you should stress this - the January 2005 event was followed by a GLE, but the SPE was not the strongest on record by far, see Figure 1.*
*Page 11, line 31: "natural variability" – you mean "random variability". The SPE is "natural".*

Modified accordingly.

*Page 11, line 32: "We encourage further research ..." that raises the question why you don't do it yourself ... maybe better "We note that further research is necessary, but outside the scope of this paper"*

The sentence is rephrased to 'We note that further research on January 2005 SPE case is necessary, to solidly confirm the EPP/dynamical/chemical factors that led to ozone destruction below 35 km, but outside the scope of this paper'.

**Anonymous referee #2**

*The manuscript is generally improved and the conclusions are given more carefully in line with the limitations of the model set up. I recommend that the final version is published after the following minor revisions are taken into account.*

*Minor revisions:*

*General: throughout the entire manuscript: xkm change to x km*
*Page 1, line 18: Despite… change to Despite that*
*Page 2, line 27: Replace comma with period and remove ''and". The publication year are missing for Denton et al. here and throughout the manuscript.*
*Page 6, line 6: Add number of winter events in the brackets*

Modified accordingly. Thank you.

*Page 9, line 26-28: rephrase the latter part of the sentence as the verbs changes from past tense to present tense. And I assume you are referring to an interpretation "are seasonal changes" should be change to "were attributed to seasonal changes"*

Modified. Thank you.

*Page 10, line 12: "extremely unique" remove "extremely"*

The word 'extremely' is now removed.

*Page 11, line 21: Start with: " In the WACCM-D model, "*

Thank you for the comment. Despite that the time range is from 1989, the fact that "robust lower stratospheric ozone decrease after SPEs was observed only once in ozone anomaly" is not in the model, but also in the observation.

**The modified manuscript is as follows, with red color marking the changes made for the revised submission.**

[revised manuscript text omitted]